The affective profiles in the USA: happiness, depression, life satisfaction, and happiness-increasing strategies

Schütz Erica 1 2 3
Sailer Uta 1 3
Al Nima Ali 1 3
Rosenberg Patricia 3
Andersson Arntén Ann-Christine 1 3
Archer Trevor 1 2 3
Garcia Danilo 3 4 5 danilo.garcia@neuro.gu.se danilo.garcia@euromail.se
1 Department of Psychology, University of Gothenburg , Gothenburg , Sweden
2 Department of Psychology, Linneaus University , Kalmar , Sweden
3 Network for Empowerment and Well-Being , Sweden
4 Center for Ethics, Law and Mental Health (CELAM), University of Gothenburg , Gothenburg , Sweden
5 Institute of Neuroscience and Physiology, The Sahlgrenska Academy, University of Gothenburg , Gothenburg , Sweden
Kronfol Ziad
Electronic publication date: 2013 Sep 10
Publication date: 2013
Volume: 1
Electronic Location ID: e156
Received 2013 Jul 19; Accepted 2013 Aug 18
Copyright: © 2013 Schtz et al.
Copyright year: 2013
Copyright holder: Schtz et al.
License: This is an open access article distributed under the terms of the Creative Commons Attribution License, which permits unrestricted use, distribution, and reproduction in any medium, provided the original author and source are credited.
License URL: https://creativecommons.org/licenses/by/3.0/

Keywords: Life satisfaction, Affective profiles, Happiness-increasing strategies, Negative affect, Happiness, Depression, Subjective well-being, Positive affect

Funding: There are no funding sources for this work.

==============================
Background. The affective profiles model categorizes individuals as self-fulfilling (high positive affect, low negative affect), high affective (high positive affect, high negative affect), low affective (low positive affect, low negative affect), and self-destructive (low positive affect, high negative affect). The model has been used extensively among Swedes to discern differences between profiles regarding happiness, depression, and also life satisfaction. The aim of the present study was to investigate such differences in a sample of residents of the USA. The study also investigated differences between profiles with regard to happiness-increasing strategies.

Methods. In Study I, 900 participants reported affect (Positive Affect Negative Affect Schedule; PANAS) and happiness (Happiness-Depression Scale). In Study II, 500 participants self-reported affect (PANAS), life satisfaction (Satisfaction With Life Scale), and how often they used specific strategies to increase their own happiness (Happiness-Increasing Strategies Scales).

Results. The results showed that, compared to the other profiles, self-fulfilling individuals were less depressed, happier, and more satisfied with their lives. Nevertheless, self-destructive individuals were more depressed, unhappier, and less satisfied than all other profiles. The self-fulfilling individuals tended to use strategies related to agentic (e.g., instrumental goal-pursuit), communal (e.g., social affiliation), and spiritual (e.g., religion) values when pursuing happiness.

Conclusion. These differences suggest that promoting positive emotions can positively influence a depressive-to-happy state as well as increasing life satisfaction. Moreover, the present study shows that pursuing happiness through strategies guided by agency, communion, and spirituality is related to a self-fulfilling experience described as high positive affect and low negative affect.

Introduction

Besides being markers of well-being (Diener, 1984), positive (PA) and negative affect (NA) have been shown to reflect stable emotional-temperamental dispositions (e.g., Watson & Clark, 1994; Tellegen, 1993). Larsen & Ketelaar (1991), for example, showed that individuals who experience high levels of PA, compared to individuals who experience low levels of PA, attend and react more intensely to positive stimuli. Likewise, individuals who experience high levels of NA, compared to individuals who experience low levels of NA, attend and react more intensely to negative stimuli (see also Norris, Larsen & Cacioppo, 2007; Lucas, 2008; Lucas & Diener, 2008). One of the most used instruments to measure affect is the Positive Affect and Negative Affect Schedule (PANAS; Watson, Clark & Tellegen, 1988) which was developed on the idea that PA and NA represents two orthogonal independent dimensions: high PA versus low PA and high NA versus low NA (see also Watson & Tellegen, 1985).

Presenting affect as being composed of two systems, each one of them categorized as high and low, leads to four different combinations beyond the two-system approach (Garcia, 2011; for a point of view on two-system theories see Keren & Schul, 2009). In this line of thinking, Archer and colleagues have developed the affective profile model through self-reported affectivity using the PANAS, generating four different profiles: self-fulfilling (high PA, low NA); high affective (high PA, high NA); low affective (low PA, low NA); and self-destructive (low PA, high NA) (e.g., Norlander, Bood & Archer, 2002; Bood, Archer & Norlander, 2004; Norlander, Johansson & Bood, 2005; Archer et al., 2007; Karlsson & Archer, 2007; Palomo et al., 2007; Palomo et al., 2008; Archer, Adolfsson & Karlsson, 2008).

Self-fulfilling individuals report feeling more energetic and optimistic than the other three affective profiles (Archer et al., 2007), while all four profiles react differently to stress and have different exercise habits and blood pressure. Self-fulfilling and high affective individuals show the best performance during stress, have a more active life, and lower blood pressure than individuals with low affective and self-destructive profiles (Norlander, Bood & Archer, 2002; Norlander, Johansson & Bood, 2005). Nevertheless, Kunst (2011) showed that high affective profiles, as self-destructive profiles, were strongly associated with increased posttraumatic stress disorder symptoms severity (for similar results among psychiatric patients see Zöller & Archer, 2009; Zöller, Karlsson & Archer, 2009). Moreover, while low affective profiles have responded maladaptively to induced stress, compared to self-fulfilling and high affective individuals (Norlander, Bood & Archer, 2002), they have at the same time reported less stress in their life, as the self-fulfilling profiles (Norlander, Johansson & Bood, 2005). Thus, the affective profile model offers something unique over and above the single dimensional framework by taking into account how both dimension interact. These interactions can be used to investigate individual differences in cognitive and emotional aspects of health and well-being (for a review see Garcia et al., 2013a).

Most past research using the affective profile model among adults has focused on measures of ill-being. Nevertheless, other important aspects of mental health are positive measures of well-being (Cloninger, 2004; Cloninger, 2006). Happiness, for example, can be usefully understood as the opposite end of depression (e.g., Joseph et al., 2004; Russell & Feldman Barrett, 1999; Watson et al., 1999; Yik, Russell & Feldman Barrett, 1999). Life satisfaction, another positive measure of well-being (Diener, 1984; Bradburn, 1969; Diener et al., 1999; Pavot, 2008), refers instead to a comparison process in which individuals assess the quality of their lives on the basis of their own self-imposed standard (Pavot & Diener, 1993).

Recent studies among adolescents have, indeed, focused on differences among profiles with respect to different measures of well-being, personality and self-regulation (e.g., Garcia & Siddiqui, 2009a; Garcia & Siddiqui, 2009b; Garcia et al., 2010; Garcia, 2009; Garcia, 2011; Garcia, 2012a; Garcia & Archer, 2012; Garcia et al., 2012; Garcia, 2013). Garcia and colleagues have found that, compared to the other profiles, individuals with a self-fulfilling profile reported higher satisfaction with life, higher psychological well-being, lower depressive symptoms, and scored higher in personality traits related to agentic values (i.e., autonomy, responsibility, self-acceptance, intern locus of control, self-control). Nevertheless, low affective individuals, compared to self-destructives, have reported being more satisfied with life and experiencing higher levels of psychological well-being (e.g., Garcia & Siddiqui, 2009b). These specific findings are also in line with the observations that low affectives and self-fulfilling individuals, report less stress in their lives (Norlander, Johansson & Bood, 2005). Moreover, Garcia (2012a) also showed that high affective and self-destructive profiles, compared to self-fulfilling and low affectives, scored higher on Neuroticism. This is not surprising, because Neuroticism is almost synonymous with negative affectivity (Watson & Clark, 1984; Tellegen, 1985)—both the high affective and self-destructive profiles have high levels of NA as a common characteristic, PA is instead suggested as strongly positively related to Extraversion. These differences in personality and well-being among profiles are suggested to, in accordance to Higgins’ (1997) prevention and promotion focus principles, result in differences in how individuals increase their happiness levels (i.e., by approaching happiness or avoiding unhappiness). Although there are current measures of happiness-increasing strategies, this suggestion has not been investigated in earlier research using the affective profiles model.

Happiness-increasing strategies

In order to intentionally pursue happiness, people seem to use different strategies. Tkach & Lyubomirsky (2006) have identified, using an open-ended survey first, 53 happiness-increasing strategies used by residents of the USA (for studies using this scale among Swedes see, Garcia, 2012b; Nima, Archer & Garcia, 2012; Nima, Archer & Garcia, 2013). Tkach & Lyubomirsky (2006) found, using factor analysis, eight factors: Social Affiliation (e.g., “Support and encourage friends”), Partying and Clubbing (e.g., “Drink alcohol”), Mental Control (e.g., “Try not to think about being unhappy”), Instrumental Goal Pursuit (e.g., “Study”), Passive Leisure (e.g., “Surf the internet”), Active Leisure (e.g., “Exercise”), Religion (e.g., “Seek support from faith”) and Direct Attempts (e.g., “Act happy/smile, etc.”).

Results have shown that these happiness-increasing strategies accounted for 52% of the variance in happiness, while the Big Five personality traits, which traditionally have been linked to happiness, accounted for 46%. Further, even after controlling for the contribution of personality, the happiness-increasing strategies accounted for 16% of the variance in happiness. However, these relationships might not be a direct one. For example, Extraversion, which is strongly related to high PA (Larsen & Ketelaar, 1991), is related to the use of the Social Affiliation strategy, which, in turn, is related to happiness. Tkach & Lyubomirsky (2006) suggested that the efficacy of the happiness-increasing strategies is also likely to vary to some extent. However, the strategy that was the most robust predictor of low levels of happiness was Mental Control, which was closely related to Neuroticism. This strategy is defined as ambivalent intentional efforts aimed, on one side and avoidance of negative thoughts and feelings as well as proneness towards contemplation of negative aspects of life on the other. Regarding the affective profiles, if the profiles differ in the way they pursue happiness (i.e., approaching happy experiences versus preventing unhappy experiences), then it could be expected that the profiles differ in the use of the strategies described here. For example, it could be expected that high PA profiles should score higher in strategies such as Social Affiliation and Active Leisure due to the close positive relationship between Extraversion and PA. High NA profiles could be expected to score higher in strategies such as Mental Control, because the positive relationship between Neuroticism and NA.

The present study

To the best of our knowledge, the affective profiles model has been mostly used among Swedish participants. Some cultures explain the world as good and controllable and others emphasize negative emotions as normal (Myers & Diener, 1995; Diener, Suh & Oishi, 1997). In this context, it is interesting to notice that the right to pursuit individual happiness is listed as an absolute right in the United States of America’s Declaration of Independence (Tkach & Lyubomirsky, 2006). The model, however, has shown identical results in the few studies using other populations (for three studies using Dutch, Indonesian, and Iranian, participants, respectively see Kunst, 2011; Adrianson et al., 2013; Garcia & Moradi, 2013).

The aim of the present study was to investigate differences in happiness, depression, life satisfaction and use of strategies to increase happiness among affective profiles in residents of the United States of America (US residents).

Study I

Method

Ethics statement

This research protocol was approved by the Ethics Committee of the University of Gothenburg and written informed consent was obtained from all the study participants.

Participants and procedure

The participants (N = 900, age mean = 28.72 sd. = 19.10, 550 males and 350 females) were US residents recruited through Amazons’ Mechanical Turk (MTurk; https://www.mturk.com/mturk/welcome). MTurk allows data collectors to recruit participants (workers) online for completing different tasks in change for wages. This method for data collection online has become more common during recent years and it is an empirical tested valid tool for conducting research in the social sciences (see Buhrmester, Kwang & Gosling, 2011). Participants were recruited by the following criteria: US resident and ability to speak and write fluently in English. Participants were paid a wage of two American dollars for completing the task and informed that the study was confidential and voluntary. The participants were presented with a battery of self-reports comprising the affect and happiness measures, as well as questions pertaining to age and gender.

Instruments

Positive Affect and Negative Affect Schedule (PANAS; Watson, Clark & Tellegen, 1988). The PANAS instructs participants to rate to what extent they generally have experienced 20 different feelings or emotions (10 PA and 10 NA) during the previous weeks, using a 5-point Likert scale (1 = very slightly, 5 = extremely). The 10-item PA scale includes adjectives such as strong, proud, and interested. The 10-item NA scale includes adjectives such as afraid, ashamed and nervous. Cronbach’s α were .87 for PA and .89 for NA in the present study.

The Short Depression-Happiness Scale (Joseph et al., 2004). This instrument consists of six items, three items measuring happiness (e.g., “I felt happy”) and three reverse-coded items measuring depressive states (e.g., “I felt my life was meaningless”). Participants rate how frequently they feel the way described in the item on a four-point scale: “never”, “rarely”, “sometimes”, “often”. In the present study, Cronbach’s α was .85 for the happiness scale and .76 for the depression scale.

Statistical treatment

We used participants’ self-reported affect measured by the PANAS from both Study I and II (N = 1,400) in order to classify participants in the four affective profiles. Participants’ PA and NA scores were divided into high and low (cut-off points: low PA = 3.0 or less; high PA = 3.1 or above; low NA = 1.8 or less; and high NA = 1.9 or above).

For Study I, the two independent variables of the study were gender and affective profiles: self-fulfilling (n = 241; 153 males, 88 females), low affective (n = 236; 137 males, 99 females), high affective (n = 180; 115 males, 65 females), and self-destructive (n = 243; 145 males, 98 females). The dependent variables were PA, NA, happiness, and depression.

Results and discussion

A Multiple Analysis of Variance (MANOVA) indicated a significant effect for gender (F(4, 889) = 4.32; p = .002, Eta2 = 0.02, power = 0.93) as well as for affective profile (F(12, 2673) = 162.19; p < .001, Eta2 = 0.42, power = 1.00). The interaction of gender and affective profile was not significant (p = .236). A between-subjects ANOVA showed significant gender effects for happiness (F(1, 892) = 7.60; p = 0.006), whereby the female participants expressed a higher level of happiness (M = 9.66, SD = 2.13) than the male participants (M = 9.35, SD = 2.33).

A between-subjects ANOVA indicated significant affective profile effects for PA (F(3, 892) = 513.78; p < .001), NA (F(3, 892) = 503.58; p < .001), happiness (F(3, 892) = 68.20; p < .001), and depression (F(3, 892) = 71.50; p < .001). A Bonferroni correction to the alpha level of .01 showed that the self-destructive group had significantly higher scores in NA and depression as well as lower scores in happiness in comparison to the other affective profiles. The self-fulfilling group differed significantly from the self-destructive profiles in all measured variables: PA, NA, happiness and depression. As expected, the high affective ones differed significantly from the self-fulfilling group in all variables except PA and the low affective ones differed significantly from the self-fulfilling group in all variables except NA. Which is not so strange since both the self-fulfilling group and the high affective group are characterized as high in PA and the same goes for self-fulfilling individuals and low affective individuals who are characterized by low NA. For further details, see Table 1.

Table 1 Mean scores in PA, NA, happiness and depression for each affective profile in Study I.

	Self-fulfilling
n = 241	High affective
n = 180	Low affective
n = 236	Self-destructive
n = 243	
Positive Affect	3.66 ± 0.44c,d	3.59 ± 0.41c,d	2.37 ± 0.52a,b	2.36 ± 0.50a,b	
Negative Affect	1.27 ± 0.21b,d	2.20 ± 0.51a,d	1.24 ± 0.21b,d	2.45 ± 0.61a,b,c	
Happiness	10.65 ± 1.77b,c,d	10.02 ± 1.94a,c,d	9.37 ± 2.22a,b,d	7.99 ± 2.12a,b,c	
Depression	4.80 ± 1.75b,c,d	5.92 ± 1.85a,d	5.75 ± 2.21a,d	7.57 ± 2.35a,b,c	
Notes.

Values represent mean scores ± SD. p < 0.01.

a Bonferroni test: compared to self-fulfilling.

b Bonferroni test: compared to the high affective.

c Bonferroni test: compared to the low affective.

d Bonferroni test: compared to the self-destructive.

Study II

Method

Participants and procedure

As in Study I, participants (N = 500, age mean = 34.08 sd. = 12.55; 217 male and 283 female) were recruited from MTurk by the following criteria: US resident and ability to both speak and write fluently in English. Participants were paid a wage of two American dollars for completing the task and informed that the study was confidential and voluntary. The participants were presented with a battery of self-reports comprising the affect, life satisfaction, and happiness-increasing strategies measures, as well as questions pertaining to age and gender.

Instruments

The same instrument as in Study I was used in Study II to measure PA and NA (i.e., the PANAS). Cronbach’s α were .88 for PA and .90 for NA in Study II.

Satisfaction with Life Scale (Diener et al., 1985). The instrument consists of 5 statements (e.g., “In most of my ways my life is close to my ideal”) for which participants are asked to indicate degree of agreement in a 7-point Likert scale (1 = strongly disagree, 7 = strongly agree). The life satisfaction score was established by summarizing the 5 statements for each participant. Cronbach’s α were .90 in the present study.

Happiness-Increasing Strategies Scales (Tkach & Lyubomirsky, 2006). In the present study, participants were asked to rate (1 = never, 7 = all the time) how often they used the strategies identified by Tkach & Lyubomirsky (2006). The happiness-increasing strategies are organized in eight clusters: Social Affiliation (e.g., “Support and encourage friends”; Cronbach’s α = 0.79), Partying and Clubbing (e.g., “Drink alcohol”; Cronbach’s α = 0.74), Mental Control (e.g., “Try not to think about being unhappy”; Cronbach’s α = 0.43), Instrumental Goal Pursuit (e.g., “Study”; Cronbach’s α = 0.76), Passive Leisure (e.g., “Surf the internet”; Cronbach’s α = 0.63), Active Leisure (e.g., “Exercise”; Cronbach’s α = 0.65), Religion (e.g., “Seek support from faith”; Cronbach’s α = 0.70), and Direct Attempts (e.g., “Act happy/smile, etc.”; Cronbach’s α = 0.56).

Statistical treatment

As detailed in Study I, both samples were used in the classification of the four affective profiles. The number of participants in each profile for Study II were as follows: 158 self-fulfilling (75 males, 83 females), 92 low affective (42 males, 50 females), 123 high affective (54 males, 69 females), and 127 self-destructive (46 males, 81 females). The affective profiles and gender were the independent variables; PA, NA, life satisfaction, and the happiness-increasing strategies were the dependent variables. An important observation here is the gender distribution between profiles. For example, in Study I there were more self-destructive males than females, while in Study II there were more self-destructive females than males. This difference might mirror the gender distribution across Study I (550 males and 350 females) and Study II (217 male and 283 females). Across both samples of females, the prevalence of the self-destructive profile was 28%, while among men was 25%. The prevalence of this profile reported here among males and females is the same that was observed among Swedes (E Schütz et al., unpublished data).

Results and discussion

First a MANOVA (3 × 2 factorial design) was applied with affective profiles and gender as independent variables and with PA, NA and life satisfaction as dependent variables. The analysis did not indicate any significant interaction effect (p = 0.14), but did indicate a significant effect for gender (F(3, 490) = 4.91; p < 0.01, Eta2 = 0.03, power = 0.91) as well as for affective profiles (F(9, 1476) = 119.15; p < 0.001, Eta2 = 0.42, power = 1.00). Secondly, a MANOVA (1 × 2 factorial design) was applied with affective profiles and gender as independent variables and with happiness-increasing strategies as dependent variables. The analysis did not indicate any significant interaction effect (p = 0.93), but did indicate a significant effect for gender (F(8, 485) = 5.85; p < 0.001, Eta2 = 0.09, power = 1.00) as well as for affective profiles (F(24, 1461) = 8.64; p < 0.001, Eta2 = 0.12, power = 1.00).

A between-subjects ANOVA was conducted in order to test gender differences in PA, NA and life satisfaction. The result indicated significant gender effects for: NA (F(1, 492) = 10.89; p < 0.01), whereby the female participants expressed a higher level of NA (M = 1.94, SD = 0.83) than the male participants (M = 1.72, SD = 0.67). This specific result stands in contrast to the results from Study I, which showed that females reported higher happiness than males. Nevertheless, this is a well-known paradox in the literature—females seem to experience positive and negative emotions equally intensively, explaining why female often report both experiencing more negative moods and depressive symptoms and also higher levels of happiness than males (Fujita, Diener & Sandvik, 1991). A between-subjects ANOVA was conducted to investigate gender differences in happiness-increasing strategies. The result indicated significant gender effects for: Social Affiliation (F(1, 492) = 17.67; p < 0.001), the female participants expressed a higher level of Social Affiliation (M = 3.43, SD = 0.56) than the male participants (M = 3.27, SD = 0.65); Instrumental Goal Pursuit (F(1, 492) = 6.60; p < 0.01), the female participants expressed a higher level of Instrumental Goal Pursuit (M = 3.33, SD = 0.81) than the male participants (M = 3.19, SD = 0.82); Religion (F(1, 492) = 23.18; p < 0.001), the female participants expressed a higher level of Religion (M = 3.08, SD = 1.13) than the male participants (M = 2.63, SD = 1.04); Passive Leisure (F(1, 492) = 9.25; p < 0.01), the female participants expressed a higher level of Passive Leisure (M = 3.30, SD = 0.55) than the male participants (M = 3.16, SD = 0.60); Direct Attempts (F(1, 492) = 4.06; p < 0.05), the female participants expressed a higher level of Direct Attempts (M = 3.66, SD = 0.58) than the male participants (M = 3.60, SD = 0.64). The differences presented here are a replication of the original study conducted by Tkach & Lyubomirsky (2006): females focus on behaviour such as maintaining relationships (i.e., Social Affiliation), pursuing career goals (i.e., Instrumental Goal Pursuit), performing religious activities (i.e., Religion), and watching TV (i.e., Passive Leisure) more frequently than males when they try to increase their happiness. As suggested by Tkach & Lyubomirsky (2006, pp. 214), the gender differences replicated here “are consistent with the gender differences reported for behaviors used to combat bad moods (Thayer, Newman & McClain, 1994)”.

In order to test differences in life satisfaction for each of the four affective profiles a between-subjects ANOVA was conducted The result indicated significant effects for life satisfaction (F(3, 492) = 49.26; p < 0.001). Further, a between-subjects ANOVA was conducted in order to test differences in happiness-increasing strategies for each of the four affective profiles. The mean scores of life satisfaction as well as for happiness-increasing strategies for all four affective profiles are presented in Table 2.

Table 2 Means in life satisfaction and happiness-increasing strategies among affective profiles in Study II.

	Self-fulfilling
N = 158	High affective
N = 123	Low affective
N = 92	Self-destructive
N = 127	
Positive Affect	3.76 ± 0.49	3.59 ± 0.42	2.44 ± 0.52	2.30 ± 0.51	
Negative Affect	1.25 ± 0.21	2.37 ± 0.58	1.20 ± 0.21	2.53 ± 0.67	
Life Satisfaction	5.17 ± 1.24	4.11 ± 1.32	4.42 ± 1.51	3.15 ± 1.49	
Social Affiliation	3.56 ± 0.53	3.51 ± 0.42	3.28 ± 0.63	3.02 ± 0.67	
Partying and Clubbing	2.12 ± 0.71	2.29 ± 0.75	2.18 ± 0.67	2.00 ± 0.64	
Mental Control	2.12 ± 0.47	2.43 ± 0.50	2.20 ± 0.42	2.59 ± 0.49	
Instrumental Goal Pursuit	3.47 ± 0.77	3.51 ± 0.67	3.07 ± 0.85	2.92 ± 0.83	
Religion	3.11 ± 1.19	2.94 ± 1.04	2.88 ± 1.11	2.57 ± 1.02	
Passive Leisure	3.22 ± 0.56	3.38 ± 0.51	3.17 ± 0.60	3.17 ± 0.63	
Active Leisure	3.39 ± 0.54	3.28 ± 0.55	3.10 ± 0.65	2.90 ± 0.65	
Direct Attempts	3.91 ± 0.50	3.68 ± 0.49	3.60 ± 0.60	3.27 ± 0.64	

A Bonferroni test, with alpha level set to .01, was conducted to compare the mean differences in life satisfaction as well as for happiness-increasing strategies between affective profiles. The results showed, replicating earlier findings, among Swedes, that the self-destructive group had lower scores in life satisfaction compared to all the other affective profiles. The self-fulfilling group had higher scores in life satisfaction compared to all the other affective profiles. Regarding happiness-increasing strategies, the results showed that the self-destructive group had lower scores in all happiness-increasing strategies except for Mental Control. For further details, see Table 3.

Table 3 Mean differences in life satisfaction and happiness-increasing strategies between affective profiles.

Affective profiles	Self-fulfilling
N = 158	High affective
N = 123	Low affective
N = 92	Self-destructive
N = 127	
Self-fulfilling					
Positive Affect		0.17*	1.32*	1.46*	
Negative Affect		−1.12*	0.05 ns	−1.28*	
Life Satisfaction		1.05*	0.75*	2.01*	
Social Affiliation		0.05 ns	0.28*	0.54*	
Partying and Clubbing		−0.16 ns	−0.06 ns	0.12 ns	
Mental Control		−0.31*	−0.09 ns	−0.47*	
Instrumental Goal Pursuit		−0.04 ns	0.39*	0.54*	
Religion		0.17 ns	0.23 ns	0.54*	
Passive Leisure		−0.16 ns	0.05 ns	0.05 ns	
Active Leisure		0.11 ns	0.29*	0.49*	
Direct Attempts		0.24*	0.31*	0.64*	
					
High affective					
Positive Affect	−0.17*		1.15*	1.29*	
Negative Affect	1.11*		−1.17*	−0.16 ns	
Life Satisfaction	−1.05*		−0.31 ns	0.96*	
Social Affiliation	−0.05 ns		0.24 ns	0.50*	
Partying and Clubbing	0.16 ns		0.11 ns	0.29*	
Mental Control	0.31*		0.23*	−0.16 ns	
Instrumental Goal Pursuit	0.04 ns		0.43*	0.58*	
Religion	−0.17 ns		0.05 ns	0.36 ns	
Passive Leisure	0.16 ns		0.21 ns	0.20 ns	
Active Leisure	−0.11 ns		0.18 ns	0.38*	
Direct Attempts	−0.23*		0.07 ns	0.40 ns	
					
Low affective					
Positive Affect	−1.32*	−1.15*		0.14 ns	
Negative Affect	−0.05	−1.17*		−1.32*	
Life Satisfaction	−0.75*	0.31 ns		1.26*	
Social Affiliation	−0.28*	−0.24 ns		0.26*	
Partying and Clubbing	0.06 ns	−0.11 ns		0.18 ns	
Mental Control	0.09 ns	−0.23*		−0.40*	
Instrumental Goal Pursuit	−0.39*	−0.43*		0.15 ns	
Religion	−0.23 ns	−0.05 ns		0.31 ns	
Passive Leisure	−0.05 ns	−0.21 ns		−0.00 ns	
Active Leisure	−0.29*	−0.18 ns		0.20 ns	
Direct Attempts	−0.31*	−0.07 ns		0.33*	
					
Self-destructive					
Positive Affect	−1.46*	−1.29*	−0.14*		
Negative Affect	1.28*	0.16 ns	1.33*		
Life Satisfaction	−2.01*	−0.96*	−1.26*		
Social Affiliation	−0.54*	−0.50*	−0.26*		
Partying and Clubbing	−0.12 ns	−0.29*	−0.18 ns		
Mental Control	0.47*	0.16 ns	0.39*		
Instrumental Goal Pursuit	−0.54*	−0.58*	−0.15 ns		
Religion	−0.54*	−0.36 ns	−0.31 ns		
Passive Leisure	−0.05 ns	−0.20 ns	0.00 ns		
Active Leisure	−0.49*	−0.38*	−0.20 ns		
Direct Attempts	−0.64*	−0.40*	−0.33*		
Notes.

ns = non significant.

* p < 0.01 with Bonferroni correction.

General discussion

The aim of this set of studies was to examine the connections between the four types of affective profiles (self-fulfilling, high affective, low affective, self-destructive) to happiness and depression (Study I), satisfaction with life and happiness-increasing strategies (Study II) in US residents. The results showed that the self-fulfilling group reported a significantly higher level of happiness and a significantly lower level of depression than all the three other groups (high affective, low affective, self-destructive). Furthermore, the self-destructive group reported a significantly higher level of depression and lower level of happiness than all the other three groups (self-fulfilling, high and low affective). The results also show that the high affective and low affective reported higher level of happiness and lower level of depression than the self-destructive group. But at the same time these groups (high and low affective) also showed significantly lower levels of happiness and significantly higher levels depression than the self-fulfilling group. As suggested by Garcia (2011), low PA among low affectives seems to influence happiness negatively as high NA influences happiness negatively among high affectives. The results presented here are corresponding to the results found in research with Swedish populations showing that high PA is related to less stress, depression, and anxiety (e.g., Garcia et al., 2012; Lindahl & Archer, 2013; Nima et al., in press). Moreover, self-fulfilling, high affective and low affective participants all have higher life satisfaction compared with self-destructive participants. This result also replicates findings among Swedish pupils where self-fulfilling, high and low affective participants showed higher level of life satisfaction compared with self-destructives (e.g., Garcia & Archer, 2012). As suggested by Lindahl & Archer (2013; see also Archer & Kostrzewa, 2013; Archer et al., 2013), positive affect might serve as an anti-depressive factor and, as suggested here, also as a protective factor for happiness and life satisfaction.

The self-fulfilling participants showed significantly higher results than all other profiles on the direct attempts strategy, suggesting that in order to increase their happiness the self-fulfilling individuals are more prone to directly attempt to smile, get themselves in a happy mood, improve their social skills, and work on their self-control. Indeed, Garcia (2012a) showed that self-fulfilling individuals score higher in personality traits related to agentic values (i.e., autonomy, responsibility, self-acceptance, intern locus of control, self-control) as measured by the Temperament and Character Inventory (Cloninger, Svrakic & Przybeck, 1993). Moreover, self-fulfilling individuals scored lower than high NA individuals (high affectives and self-destructives) in the strategy of mental control. The mental control scale has been defined as ambivalent behavior, that is, the individual using this happiness-increasing strategy make efforts to avoid negative experiences by suppressing negative thoughts and feelings but also ruminating about negative aspects of life (Tkach & Lyubomirsky, 2006). These tendencies may not only prolong unhappiness, suppressing negative thoughts actually may end up in maintaining these thoughts and thereby aggravate negative affect (Tkach & Lyubomirsky, 2006), which may explain why these tendencies are more frequent among high affective and self-destructive than self-fulfilling individuals.

Compared to low PA individuals (i.e., low affectives and self-destructives), the self-fulfilling individuals also reported using three of the other happiness-increasing strategies more often: social affiliation, instrumental goal pursuit, active leisure. Social affiliation activities comprise communal (i.e., cooperation) values to guide behavior such as: supporting and encouraging friends, helping others, trying to improve oneself, interacting with friends, and receiving help from friends (Tkach & Lyubomirsky, 2006). Instrumental goal pursuit includes activities directed to achieving goals by trying to reach one’s full potential, studying, organizing one’s life and goals, and striving for the accomplishment of tasks (Tkach & Lyubomirsky, 2006). Finally, the use of active leisure comprises a propensity for wellness through fitness and flow, that is, exercising and working on hobbies or activities in which the individual uses her/his strengths and becomes absorbed by the activity itself (Tkach & Lyubomirsky, 2006). In other words, both instrumental goal pursuit and active leisure comprises agentic (i.e., autonomous, self-directed) values guiding behavior in order to approach well-being. Indeed, among Swedes (Nima, Archer & Garcia, 2012; Nima, Archer & Garcia, 2013), these three strategies (social affiliation, instrumental goal pursuit, and active leisure) have been found to be positively related to subjective well-being. Agency and cooperation are also related to mental health, dysfunction and suffering (Cloninger & Zohar, 2011; Garcia, Anckarsäter & Lundström, 2013; Garcia et al., 2013b; Garcia, Nima & Archer, in press) and are suggested to help the individual become happier and healthier (Cloninger, 2013; see also Johansson et al., 2013, who showed that increases in agency and cooperation are associated to improvement in depression). Moreover, compared to the self-destructives, the self-fulfilling individuals reported more frequently seeking support from faith, performing religious activities, praying, and drinking less alcohol (i.e., the religion happiness-increasing strategy). Indeed, Cloninger (2013) has suggested that while agency and cooperation might lead to happiness and health, spiritual values might be needed for becoming a self-fulfilled individual that lives in harmony with the changing world. See Fig. 1 for a summary of the results.

Figure 1 Summary of the results from Study I and II showing the differences between affective profiles in happiness, depression, life satisfaction, and the happiness-increasing strategies.

Limitations and future research

One limitation of the present set of studies is that the results are based on MTurk workers’ self-reports. Nevertheless, consistent with earlier research suggesting MTurk as a valid tool for collecting data using personality scales (Buhrmester, Kwang & Gosling, 2011), other researchers have found that health measures using MTurk data shows satisfactory internal reliability and test-retest reliability (Shapiro, Chandler & Mueller, 2013). Furthermore, the prevalence of depression among MTurk workers matches the prevalence of this illness in the general population which makes MTurk a valid tool even for clinical research (Shapiro, Chandler & Mueller, 2013). The measures used here are validated and reliable measures of happiness, depression, life satisfaction, and affect; however, there are other established measures that could have been used for the measurement of depression (e.g., The Patient Health Questionnaire; Kroenke, Spitzer & Williams, 2001). The Short Depression-Happiness Scale (Joseph et al., 2004), used in Study I, was found appropriate firstly because it was developed as a short easy-to-distribute scale based on the increasing awareness of the therapeutic potential of the positive psychological perspective (e.g., Cloninger, 2006; Joseph & Linley, 2004; Keyes & Lopez, 2002). This scale has shown good psychometric properties of internal consistency reliability (Cronbach’s α between .77–92), test-retest reliability (r = .68 in a 2-week interval), and convergent and discriminant validity with measures of depression (Beck’s Depression Inventory), happiness (Oxford’s Happiness Inventory) and personality (NEO Five Factor Inventory) (Joseph et al., 2004).

The lack of studies in adult populations using the affective profiles model and positive measures of well-being did not permit comparison of the results presented to other than earlier research among adolescents and young adults, thus, showing the need for further studies on adults regarding these factors. The reliability coefficients for some of the happiness-increasing strategies were low (e.g., Direct Attempts showed an Cronbach’s α = .56). In studies among Swedes these scales have been modified through factor analyses (Nima, Archer & Garcia, 2013). Although most of the scales in the present study showed alphas above .63, further studies focusing in the validation of these scales are needed. Furthermore, specific emotions vary widely across the lifespan. Findings among men and women in the US, for example, show that as people age they become less stressed and angry, although worry seems to persist as a negative emotion in people’s lives during middle age (Stone et al., 2010). Positive emotions such as happiness and enjoyment along with negative emotions such as sadness, however, show very limited change with age (Stone et al., 2010). Although the present study did not aim to investigate variations in specific emotions with respect to age, further studies exploring increases/decreases in PA and NA are needed.

Finally, since median splits distort the meaning of high and low, it is plausible to criticize the validity of the procedure used here to create the different affective profiles—scores just-above and just-below the median become high and low by fiat, not by reality (Schütz, Archer & Garcia, 2013). Nevertheless, a recent study (MacDonald & Kormi-Nouri, 2013) used k-means cluster analysis to test if the affective profiles model emerged as theorized by Archer and colleagues. The affective profile model was replicated using the k-means cluster analysis and the four affective profiles emerged as the combinations of high vs. low affectivity. The procedure used by these researchers is useful for person-oriented analyses (see Bergman, Magnusson & El-Khouri, 2003), thus, suggesting the original procedure by Archer as valid.

Conclusion

The present set of studies expands earlier results among Swedes to a relative large sample of US residents. The results suggest that the affective profiles model distinguish important differences in happiness, depression, and life satisfaction between individuals. These differences suggest that promoting positive emotions can positively influence a depressive-to-happy state as well as increasing life satisfaction. Moreover, the present study describes further how affective profiles differ with regard to happiness-increasing strategies. These specific results suggest that the pursuit of happiness through agentic, communal, and spiritual values leads to a self-fulfilling experience defined as frequently experiencing positive emotions and infrequently experiencing negative emotions.

“It was right then that I started thinking about Thomas Jefferson on the Declaration of Independence and the part about our right to life, liberty, and the pursuit of happiness. And I remember thinking how did he know to put the pursuit part in there?”

Will Smith as Christopher Gardner in The Pursuit of Happyness

Additional Information and Declarations

Competing Interests

Author Contributions

Human Ethics

There are no competing interests.

Erica Schütz analyzed the data, wrote the paper.

Uta Sailer, Patricia Rosenberg and Ann-Christine Andersson Arntén wrote the paper.

Ali Al Nima conceived and designed the experiments, performed the experiments, contributed reagents/materials/analysis tools, wrote the paper.

Trevor Archer conceived and designed the experiments, wrote the paper.

Danilo Garcia conceived and designed the experiments, performed the experiments, analyzed the data, contributed reagents/materials/analysis tools, wrote the paper.

The following information was supplied relating to ethical approvals (i.e., approving body and any reference numbers):

This research protocol was approved by the Ethics Committee of the University of Gothenburg and written informed consent was obtained from all the study participants.

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
