# Peer review of "The affective profiles in the USA: happiness, depression, life satisfaction, and happiness-increasing strategies"

_PeerJ, doi:10.7717/peerj.156_

## Round 0.1 · original submission · Major Revisions

Two reviewers have submitted their comments. They both agree that the methodology is sound but that there is room for improvement in the reporting. Please use the reviewers' comments to make the manuscript more scholarly and easier to read. Please include in an accompanying letter the specific changes you have made and/or any rebuttal to the reviewers' concerns.

·

Basic reporting

(1) The unique contribution and significance of this paper are not sufficiently explained. If the U.S. population were of interest for specific reasons (other than a different setting for replication purposes), more should have been said about the U.S. population and its differences with previously examined populations (e.g., Dutch, Indonesians, Iranians). Hypotheses could have stated expected differences in outcomes based on certain characteristics of U.S. culture.

(2) The introduction is not sufficiently organized, leaving the reader somewhat confused about the main objectives/interests of the present study. Overall, the introduction could be significantly shorter. There are nearly 6 pages of introductory material, and the relevance of some parts is not clear enough.

(3) The authors present differences based on gender but do not fully explain how their findings fit into a broader literature. Furthermore, variations in mental health (including positive affect) are known to vary widely across the lifespan. For example, a recent study by Stone et al. (2010) found that for both men and women, levels of stress, anger, worry and sadness increase beginning at age 18 and takes on a u-shape, with steady decreases in well-being beginning in late adolescence. In the present study, the sample appears to have a wide age range, but the authors do not examine age.

(4) Findings from Study 1 and 2 are seemingly contradictory for some outcomes, and it will help for the authors to reconcile or discuss these differences. For example, in Study 1, there were more self-destructive males than females while in Study 2 there were more self-destructive females than males.

(5) My understanding is that there are significantly more limitations to using MTurk than the authors mention. I know that this is becoming an increasingly common practice in the social sciences, but the authors should say more about the potential limitations of their approach.

Experimental design

No Comments

Validity of the findings

No Comments

Additional comments

The comments above are all under the "Basic Reporting" category, because the methods seem reasonable but there could be improvement in how the study is framed and the findings are presented.

·

Basic reporting

The authors report data from two studies that examine the connections between the four types of affective profiles (self-fulfilling, high affective, low affective, self-destructive) to happiness and depression (Study I), satisfaction with life and happiness-increasing strategies (Study II) among US residents. Results showed that the self-fulfilling group reported a significantly higher level of happiness and a significantly lower level of depression than all the three other groups (high affective, low affective, self-destructive).

Innovative use of the MTurk to recruit US participants!

The authors are explicit with their aim about generalizing previous results from the Swedish general population. Reporting is clear and presented in a logical structure.

Great illustration of the results in Figure 1. It could be more clear in the description of the Figure that it illustrates the results from the present study.

Experimental design

The design is valid and seem appropriate to the research question.

Validity of the findings

The authors have included various secondary measures in the two studies. Research that support the validity of the included measure need to be reported. Best would be previously published data that support this, and not just for the present sample. For example, the authors have used the "Short Depression-Happiness Scale" which is a measure that the reviewer have not heard of. The authors need to argue for including such measure and not an established measure (e.g. the PHQ-9). Probably, the authors will need to include this as a limitation.

Additional comments

Some minor comments:

* The last sentence in the Conclusion ("Showing that agentic…") would benefit from rewriting.

* The "Cloninger, C. R., & Zohar, A. H." reference is misplaced in the reference list

---

## Round 0.2 · accepted · Accept

The authors have addressed all major concerns raised by the reviewers. The manuscript in its present format is acceptable for publication. It adds a new dimension to the literature on the affective profiles model.

·

Basic reporting

none

Experimental design

None

Validity of the findings

None

Additional comments

The revisions are adequate and I have no further comments.